# LOCAL PERMUTATION EQUIVARIANCE FOR GRAPH NEURAL NETWORKS

## ABSTRACT

In this work we develop a new method, named *locally permutation-equivariant graph neural networks*, which provides a framework for building graph neural networks that operate on local node neighbourhoods, through sub-graphs, while using permutation equivariant update functions. Message passing neural networks have been shown to be limited in their expressive power and recent approaches to over come this either lack scalability or require structural information to be encoded into the feature space. The general framework presented here overcomes the scalability issues associated with global permutation equivariance by operating on sub-graphs through restricted representations. In addition, we prove that there is no loss of expressivity by using restricted representations. Furthermore, the proposed framework only requires a choice of $k$-hops for creating sub-graphs and a choice of representation space to be used for each layer, which makes the method easily applicable across a range of graph based domains. We experimentally validate the method on a range of graph benchmark classification tasks, demonstrating either state-of-the-art results or very competitive results on all benchmarks. Further, we demonstrate that the use of local update functions offers a significant improvement in GPU memory over global methods.

## 1 INTRODUCTION

Many forms of data are naturally structured as graphs such as molecules, bioinformatics, social, or financial and it is therefore of interest to have algorithms which operate over graphs. Machine learning on graphs has received much interest in recent years, with the general framework of a message passing network providing both a useful inductive bias and scalability across a range of domains (Gilmer et al., 2017). However, Xu et al. (2019) show that a model based on a message passing framework with permutation invariant aggregation functions is limited in expressive power. Therefore, there exists many non isomorphic graphs that a model of this form cannot distinguish between. Figure 5 demonstrates two non-isomorphic graphs for which a message passing framework with max pooling would not be able to distinguish between the two graphs.

More expressive graph networks exists and a common measure of expressivity is the Weisfeiler-Lehman (WL) test. One such method of building more expressive networks is based directly on the WL test. Here a neural network architecture is built based on variants of the WL test. Bouritsas et al. (2020) use a permutation invariant local update function, but incorporate permutation equivariant structural information into the feature space. Morris et al. (2019a) build models based on different WL variants that consider local and global connections. Bodnar et al. (2021b) introduce a WL test on simplicial complexes and incorporate this into a message passing scheme. Bodnar et al. (2021a) extend work on simplicial complexes to cell complexes, which subsume simplicial complexes.

On the other hand, rather than trying to directly incorporate techniques from WL tests directly into networks other model make use of permutation symmetries to build permutation equivariant graph neural networks (Maron et al., 2018). This model can be built for $k$-order feature spaces and it was shown by Maron et al. (2019) that such models can distinguish between non-isomorphic graphs as well as the $k$-WL test. Natural graph networks are a different class of graph neural network, where the constraint placed upon the linear layer is that of naturality (de Haan et al., 2020). The naturality constraint says that for each isomorphism class a map must be chosen that is equivariant to automorphisms.

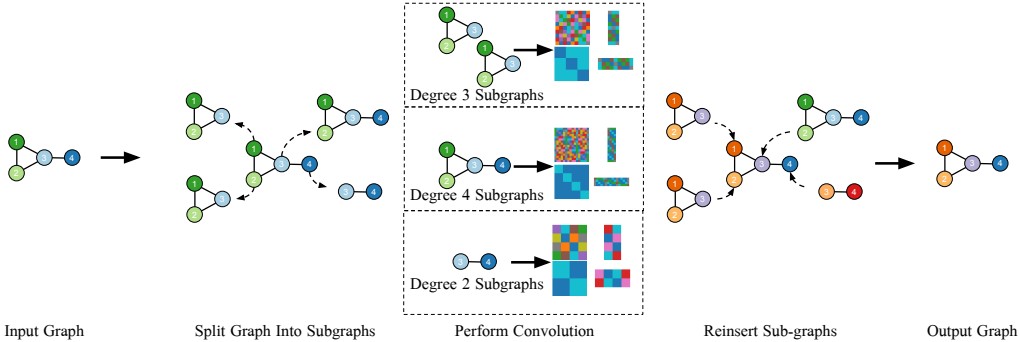

Figure 1: The architecture of a layer in a locally permutation equivariant graph network is presented. It combines a method of splitting the input graph into sub-graphs, the chosen method of weight sharing, permutation representations as update constraints, and the reconstruction of the graph. In the layer presented here the sub-graph is created by selecting nodes that are 1-hop away from the central update node. The weight sharing scheme involves sharing weights across sub-graphs of the same size. The update of node and graph features makes use of permutation representations.

In general the task of learning on graphs consists of utilising many graphs of different sizes. Current methods for utilising permutation equivariant graph neural networks require that the graph be represented as an adjacency tensor, which limits there scalability. Furthermore, global natural graph networks also perform computations on entire graph features, which leads to a large computational complexity for large graphs. Local gauge symmetries have been considered to build models with local equivariance (Cohen et al., 2019). This approach improves scalability of models by utilising local update functions, however for graphs we do not have a single local symmetry. Currently this is overcome in the majority of graph neural networks presented by utilising some form of message passing, but, in general, all works use a permutation invariant aggregation function leading to good scalability but poor expressivity. Local natural graph networks attempt to overcome the limited expressivity through placing a local naturality constraint on the message passing and having different message passing kernels on non-isomorphic edges.

Through considering graph neural networks from an elementary category theory perspective and making use of aspects of group theory we present a framework for building local permutation equivariant models. This allows us to build a graph neural network model with local update functions that are permutation equivariant by considering restricted representations of the representation space of the whole graph. Further, we prove that this does not cause a loss of expressivity of the model and that this maintains the option to have a $k$-order feature space that ensures expressivity equal to $k$-WL test. Also, by constraining the kernel space under restricted representations, a natural weight sharing scheme becomes apparent, namely sharing weights across local graph neighbourhoods of the same degree. The approach of building models with a framework based on group theory makes clear the generality of the approach, where choices of representation space can be made for each convolutional layer without requiring prior information such as structural information to be encoded into the feature space. This framework can also be shown to include other leading methods as specific cases.

## 2 BACKGROUND

### 2.1 GRAPH NETWORKS

Different graph neural networks express graphs in alternative forms. Generally, for a message passing model, a matrix of node features and a matrix of edge features is combined with a sparse edge index array specifying the connectivity of the graph. In other works, the graph is provided in a dense format, where the graph is given as a adjacency tensor with node and edge features held in one tensor. In this work we present the graph as follows:

**Definition 1** *A Concrete Graph $G$ is a finite set of nodes $\mathcal{V}(G) \subset \mathbb{N}$ and a set of edges $\mathcal{E}(G) \subset \mathcal{V}(G) \times \mathcal{V}(G)$.*

The set of node ids may be non-contiguous and we make use of this here as we extract overlapping sub-graphs when performing the local updates. The same underlying graph can be given in may forms by a permutation of the ordering of the natural numbers of the nodes.

**Definition 2** *A sub-Concrete Graph $H$ is created by taking a node $i \in \mathcal{V}(G)$, and extracting the nodes $j \in \mathcal{V}(G)$ and edges $(i,j) \subset \mathcal{V}(G) \times \mathcal{V}(G)$, such that there is a connection between nodes $i$ and $j$.*

Once a sub-concrete graph has been extracted, this same underlying sub-graph could be expressed through different permutations of the underlying numbering of the nodes. For brevity we refer to sub-concrete graphs as subgraphs throughout the paper.

**Definition 3** *A Graph isomorphism, $\phi : G \to G'$ is a bijection between the vertex sets of two graphs $G$ and $G'$, such that two vertices $u$ and $v$ are adjacent in $G$ if and only if $\phi(u)$ and $\phi(v)$ are adjacent in $G'$. This mapping is edge preserving, i.e. satisfies for all $(i,j) \in \mathcal{V}(G) \times \mathcal{V}(G)$:*

$$(i,j) \in \mathcal{E}(G) \iff (\phi(i), \phi(j)) \in \mathcal{E}(G')$$

*An isomorphism from the graph to itself is known as an automorphism.*

Relabelling of the graph by a permutation of the nodes is called a graph isomorphism, where an example of two isomorphic graphs is given in Figure 5. We desire that the linear layers of the graph neural network respect the composition of graph isomorphisms. This requires us to define the feature space of the graphs and how feature spaces of isomorphic graphs are related.

## 2.2 PERMUTATION REPRESENTATIONS

The feature space of the graphs is a vector space $V$, where a representation of the group $G$ is a homomorphism $\rho : G \to \mathrm{GL}(V)$ of $G$ to the group of automorphisms of $V$ (Fulton & Harris, 2013). A map $K_G$ between two representations of $G$ is a vector space map. The elements of the group $g \in G$ can act on a vector $v \in V$ by the representation matrix $v \to \rho(g)v$. The symmetric subspace of the representation is the space of solutions to the constraint $\forall g \in G : \rho(g)v = v$. Here we are considering the symmetries of the symmetric group $S_n$. This constraint can be solved for different order representations (Maron et al., 2018; Finzi et al., 2021). We present the space of linear layers mapping from $k$-order representations to $k'$-order representations in Figure 2. In addition, for the linear map $K_G$, we require that if a graph is passed through $K_G$ and then transformed by permutation to an isomorphic graph this result is the same as if a graph is transformed by the same permutation to an isomorphic graph and then passed through $K_G$. In short, this requires that permutation equivariance is satisfied.

## 2.3 CATEGORY THEORY

This section does not provide a complete overview of category theory, nor even a full introduction, but aims to provide a sufficient level of understanding to aid the reader with further sections of the paper, where we believe presenting the comparison between models from a category theory perspective makes more clear the distinctions between them. A category, $\mathcal{C}$, consists of a set of objects, $\mathrm{Ob}(\mathcal{C})$, and a set of morphisms (structure-preserving mappings) or arrows, $f : A \to B$, $A, B \in \mathrm{Ob}(\mathcal{C})$. There is a binary operation on morphisms called composition. Each object has an identity morphism. Categories can be constructed from given ones by constructing a subcategory, in which each object, morphism, and identity is from the original category, or by building upon a category, where objects, morphisms, and identities are inherited from the original category. A functor is a mapping from one category to another that preserves the categorical structure. For two categories $\mathcal{C}$ and $\mathcal{D}$ a functor $F : \mathcal{C} \to \mathcal{D}$ maps each object $A \in \mathrm{Ob}(\mathcal{C})$ to an object $F(A) \in \mathrm{Ob}(\mathcal{D})$ and maps each morphism $f : A \to B$ in $\mathcal{C}$ to a morphism $F(f) : F(A) \to F(B)$ in $\mathcal{D}$.

**Definition 4** *A groupoid is a category in which each morphism is invertible. A groupoid where there is only one object is usually a group.*

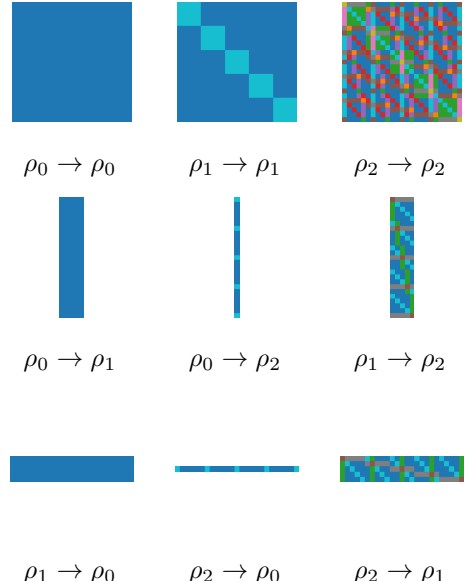

Figure 2: Bases for mappings to and from different order permutation representations, where $\rho_k$ is a $k$-order representation. Each color in a basis indicates a different parameter. $\rho_0 \rightarrow \rho_0$ is a mapping from a $0$-order representation to a $0$-order representation, i.e. a graph level label to graph level label, and has 1 learnable parameter. $\rho_1 \rightarrow \rho_1$ is a mapping from a $1$-order representation to a $1$-order representation, i.e. a node level label to node level label, and has 2 learnable parameters, one mapping node features to themselves and the other mapping node features to other nodes. Further, there are mappings between different order representation spaces and higher order representation spaces.

## 3 GLOBAL EQUIVARIANT GRAPH NETWORKS

### 3.1 GLOBAL PERMUTATION EQUIVARIANCE

Global permutation equivariant models have been considered by Hartford et al. (2018); Maron et al. (2018; 2019); Albooyeh et al. (2019), with Maron et al. (2018) demonstrating that for order-2 layers there are 15 operations that span the full basis for an permutation equivariant linear layer. These 15 basis elements are shown in Figure 2 with each basis element given by a different color in the map from representation $\rho_2 \rightarrow \rho_2$. Despite these methods, when solved for the entire basis space, having expressivity as good as the $k$-WL test, they operate on the entire graph. Operating on the entire graph features limits the scalability of the methods. In addition to poor scalability, global permutation appears to be a strong constraint to place upon the model. In the instance where the graph is flattened and an MLP is used to update node and edge features the model would have $n^4$ trainable parameters, where $n$ is the number of nodes. On the other hand, a permutation equivariant update has only 15 trainable parameters and in general $15 \ll n^4$.

Viewing a global permutation equivariant graph network from a category theory perspective there is one object with a collection of arrows representing the elements of the group. Here the arrows or morphisms go both from and to this same single object. The feature space is a functor which maps from a group representation to a vector space. For a global permutation equivariant model the same map is used for every graph.

$$\overset{g_1}{\underset{g_2 \quad e}{G}}$$

Symmetric Group

### 3.2 GLOBAL NATURALITY

Global natural graph networks (GNGN) consider the condition of naturality, (de Haan et al., 2020). GNGNs require that for each isomorphism class of graphs there is a map that is equivariant to automorphisms. This naturality constraint is given by the condition $\rho'(\phi) \circ K_G = K_{G'} \circ \rho(\phi)$, which must hold for every graph isomorphism $\phi : G \to G'$ and linear map $K_G$. While the global permutation equivariance constraint requires that all graphs be processed with the same map, global naturality allows for different, non-isomorphic, graphs to be processed by different maps and as such is a generalisation of global permutation equivariance. As is the case for global permutation equivariant models, GNGNs scale poorly as the constraint is placed over the entire graph and linear layers require global computations on the graphs.

Viewing a GNGN from a category theory perspective there is a different object for each concrete graph, which form a groupoid. Then, there is a mosphism or arrow for each graph isomorphism. These can either be automorphisms, if the arrow maps to itself, or isomorphisms if the arrow maps to a different object. The feature spaces are functors which map from this graph category to the category of vector spaces. The GNG layer is a natural transformation between such functors consisting of a different map for each non-isomorphic graph.

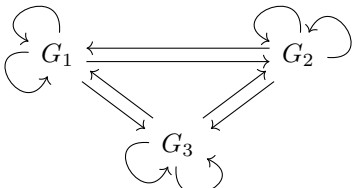

Groupoid of Concrete Graphs

## 4 LOCAL EQUIVARIANT GRAPH NETWORKS

Local equivariant models have started to receive attention following the successes of global equivariant models and local invariant models. The class of models that are based on the WL test are not in general locally permutation equivariant in that they still use a message passing model with permutation invariant update function. Despite this, many of these models inject permutation equivariant information into the feature space, which improves the expressivity of the models (Bouritsas et al., 2020; Morris et al., 2019a; Bodnar et al., 2021b;a). The information to be injected into the feature space is predetermined in these models by a choice of what structural or topological information to use, whereas our model uses representations of the permutation group, making it a very general model that still guarantees expressivity.

In contrast to utilising results from the WL test covariant compositional networks (CCN) look at permutation equivariant functions, but they do not consider the entire basis space as was considered in Maron et al. (2018) and instead consider four equivariant operations (Kondor et al., 2018). This means that the permutation equivariant linear layers are not as expressive as those used in the global permutation equivariant layers. Furthermore, in a CCN the node neighbourhood and feature dimensions grow with each layer, which can be problematic for larger graphs and limits their scalability. Another local equivariant model is that of local natural graph networks (LNGN) (de Haan et al., 2020). An LNGN uses a message passing framework, but instead of using a permutation invariant aggregation function, it specifies the constraint that node features transform under isomophisms of the node neighbourhood and that a different message passing kernel is used on non-isomorphic edges. In practice this leads to little weight sharing in graphs that are quite heterogeneous and as such the layer is re-interpreted such that a message from node $p$ to node $q$, $k_{pq}v_p$, is given by a function $k(G_{pq}, v_p)$ of the edge neighbourhood $G_{pq}$ and feature value $v_p$ at $p$.

Viewing a LNGN from a category theoretic perspective there is a groupoid of node neighbourhoods where morphisms are isomorphisms between node neighbourhoods and a groupoid of edge neighbourhoods where morphisms are ismorphisms between edge neighbourhoods. In addition, there is a functor mapping from edge neighbourhoods to the node neighbourhood of the start node and a functor mapping similarly but to the tail node of the edge neighbourhood. The node feature spaces

are functors mapping from the category of node neighbourhoods to the category of vector spaces. Further, composition of two functors creates a mapping from edge neighbourhoods to the category of vector spaces. A LNG kernel is a natural transformation between these functors.

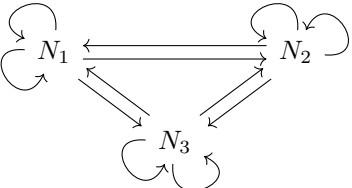

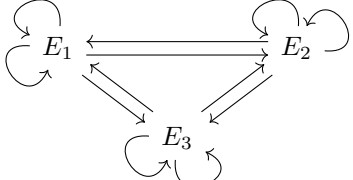

Groupoid of Node Neighbourhoods              Groupoid of Edge Neighbourhoods

## 5 LOCAL PERMUTATION EQUIVARIANCE

A local permutation equivariant graph network (LPEGN) improves upon the scalability of global permutation equivariant models by considering permutation equivariance at lower scales. Here, instead of performing the update function on the entire graph, we perform the update function on node neighbourhoods as is done in message passing models. Furthermore, while performing the update functions on node neighbourhoods, we maintain improved expressivity through using $k$-order permutation representations. The intuition behind imposing permutation equivariance on node neighbourhoods rather than the entire graph is that the model can learn expressive features about a part of the sub-graph without requiring knowledge of permutations multiple hops away from the central update node. This framework generalises global permutation equivariant models as it is compatible with all length scales, meaning that, if the graph structure is used to determine node neighbourhoods, then any $k$ value can be chosen to determine the $k$-hops from the central update node producing the sub-graph which permutation equivariance is required for. Therefore, if the value chosen for the $k$-hops is sufficiently large then the layer becomes a global permutation update. The basis functions for different order representation spaces are given with the split into different degrees for a 1-hop node neighbourhood in Figure 1. The method therefore requires a choice of $k$ for the number of hops away from the central node to consider in the local update and we discuss this choice in Section 5.2. In addition, the framework then allows for a choice of weight sharing, which we discuss in Section 5.3.

### 5.1 RESTRICTED REPRESENTATION

Given a graph comprised of $n$ nodes, global equivariant models consider the permutation representation of the permutation group $G = S_n$, namely the representation $\rho : G \to \mathrm{GL}(\mathbb{R}^c)$. Here we consider local updates on sub-graphs with $m$ nodes, where we are interested in the sub-group $H = S_m \leq S_n$. Therefore we can consider the restricted representation of the sub-group $S_m$, where the restricted representation can be seen as dropping some symmetries from the group $S_n$. The restricted representation is denoted by $\tilde{\rho} := \mathrm{Res}_H^G(\rho) : H \to \mathrm{GL}(\mathbb{R}^c)$. The global equivariance case using representations, $\rho$, and the case using restricted representations, $\tilde{\rho}$, are shown in Figure 3. Both figures show a basis mapping from order 1 to order 1 permutation representation. The restricted repre-

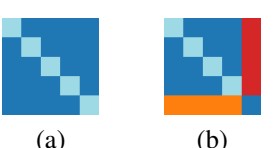

(a)              (b)

Figure 3: (a) Regular representation. (b) Restricted representation.

sentation $\mathrm{Res}_{S_4}^{S_5}$ drops the permutation symmetry associated to node 5. Dropping the permutation symmetry of node 5 results in 3 additional parameters, one for the update of node 5 based on node 5's features, another for the update of node 5 based on the features of the other nodes in the graph, and a final parameter for the update of the other nodes in the graph based on node 5's features. We proove that using restricted representations in our framework has no loss of expressivity in Appendix A.8.

## 5.2 CHOICE OF LOCAL NEIGHBOURHOOD

The LPEGN model framework performs the permutation equivariant update on local sub-graphs, although a choice can be made as to how these sub-graphs are created. One option is the use the underlying graph structure and choose a $k$ value to extract local neighbourhoods that include nodes which are at most $k$-hops from the central node. This method creates a sub-graph for each node in the graph. Here the choice of the $k$ value can be seen as choosing a length scale for which the permutation symmetry should be exploited over. In other words, choosing a value of $k = 1$ is the shortest length scale and node features will be updated such that they are permutation equivariant to their 1-hop neighbours, but not equivariant to nodes further away in the graph. On the other hand, choosing a $k$ value sufficiently large will create a model equivalent to global permutation equivariant models, where each update is permutation equivariant to permutations of the entire graph. Throughout this work we choose $k = 1$ unless otherwise stated to take the most local permutation equivariant updates. We show how this choice of $k$ value will impact the method through analysing the MUTAG dataset in Figure 10.

## 5.3 CHOICE OF WEIGHT SHARING

In general when constructing the sub-graphs a variety of different sized sub-graphs are found due to differing degrees of the nodes in the graph. This allows for a further choice, namely the weight sharing method to be used. Given that the permutation equivariance constraint is a strong constraint to place over the linear layers, we perform weight sharing across sub-graphs of the same size. This means that sub-graphs of different sizes do not share weights and can be updated differently. The intuition for this is that sub-graphs of the same size already have some similarity in that they are of the same size, while sub-graphs of a different size are less likely to be similar and hence should be updated differently. Throughout this paper we choose to use weight sharing across local neighbourhoods of the same size degree, although in situations where there is very few local neighbourhoods of a particular size we group these together.

## 5.4 CHOICE OF REPRESENTATION SPACE

In Section 5.1 we considered the restricted representation of a sub-group $S_m \leq S_n$ and in Section 5.2 we detailed how local sub-graphs are selected. Here we must make a connection between the two to present the representational space used in our LPEGN framework. When focusing in on the nodes that we didn't drop the permutation symmetry of it can be seen, in Figure 3, that for these nodes the restricted representation is equivalent to the global permutation equivariant representation. Furthermore, given our choice of sub-graph construction we would seek to drop the permutation symmetry from a node in the graph due to the fact it is not connected to the central update node. Therefore the edge features connecting the central node to the node we are dropping the permutation symmetry of are zero. Hence, we are not interested in the additional parameters introduced in the restricted representation connecting the two nodes. Furthermore, as the node we are dropping permutation symmetries for is not connected to the chosen sub-graph we are also not interested in the additional parameters introduced in the restricted representation for this node. As a result, due to the choice of sub-graph construction, the restricted representation for our sub-group has zero features in the position of new parameters introduced and is therefore equivalent to the permutation representation on a lower dimensional space. Therefore where global permutation equivariant updates use representations $\rho : G \rightarrow \mathrm{GL}(\mathbb{R}^c)$, our local permutation equivariant model uses representations $\tilde{\rho} : H \rightarrow \mathrm{GL}(\mathbb{R}^{\bar{c}})$, where $\bar{c} \leq c$. The scheme for creating representations of local neighbourhoods is shown in Figure 1, where some representations of the local neighbourhoods are shown.

## 5.5 LOCAL PERMUTATION EQUIVARIANT GRAPH NETWORK

A LPEGN combines the chosen method of creating sub-graphs as local neighbourhoods with a choice of weight sharing scheme and makes use of permutation representations on these sub-graphs. The process of creating sub-graphs, updating based on the choice of weight sharing using permutation representations, and re-constructing the graph structure is presented in Figure 1.

Viewing a LPEGN from a category theoretic perspective, each different size node neighbourhood is a sub-group, $H$, which is a different object. There are morphisms or arrows for each permutation

of the neighbourhood. This forms a groupoid. The sub-group representations are functors from the category of node neighbourhoods to the category of vector spaces.

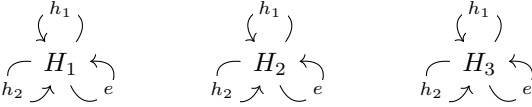

Groupoid of Symmetric Sub-Groups

# 6 EXPERIMENTS

## 6.1 GRAPH BENCHMARKS

We tested our method on a series of 7 different real-world graph classification problems from the benchmark of (Yanardag & Vishwanathan, 2015). It is noteworthy to point out some interesting features of each dataset. We note that both MUTAG and PTC are very small datasets, with MUTAG only having 18 graphs in the test set when using a 10 % testing split. Further, the Proteins dataset has the largest graphs with an average number of nodes in each graph of 39. Also, NCI1 and NCI109 are the largest datasets having over 4000 graphs each, leading to less spurious results. Finally, IMDB-B and IMDB-M generally have smaller graphs, with IMDB-M only having an average number of 13 nodes in each graph. The small size of graphs coupled with having 3 classes appears to make IMBD-M a challenging problem.

Table 1 compares our LPEGN model to a range of other methods. This highlights that our method achieves a new state-of-the-art result on the NCI1 dataset and is the second strongest on PTC and NCI109. Furthermore, our method performs competitively across all datasets. We achieve a poor ranking score on the Proteins datasets, although the classification accuracy of the model is competitive with leading results and only falls slightly short of the bulk of other methods. A comparison of the distribution of training accuracy is presented in figure 8 and a ranking based method is presented in Figure 9.

Table 1: Comparison between our LPEGN model and other deep learning methods from de Haan et al. (2020). Larger mean results are better.

| Dataset | MUTAG | PTC | PROTEINS | NCI1 | NCI109 | IMDB-B | IMDB-M |
|---|---|---|---|---|---|---|---|
| size | 188 | 344 | 1113 | 4110 | 4127 | 1000 | 1500 |
| classes | 2 | 2 | 2 | 2 | 2 | 2 | 3 |
| avg node # | 17.9 | 25.5 | 39.1 | 29.8 | 29.6 | 19.7 | 13 |
| Results | | | | | | | |
| GDCNN (Zhang et al., 2018) | 85.8±1.7 | 58.6±2.5 | 75.5±0.9 | 74.4±0.5 | NA | 70.0±0.9 | 47.8±0.9 |
| PSCN (Niepert et al., 2016) | 89.0±4.4 | 62.3±5.7 | 75±2.5 | 76.3±1.7 | NA | 71±2.3 | 45.2±2.8 |
| DCNN (Atwood & Towsley, 2016) | NA | NA | 61.3±1.6 | 56.6±1.0 | NA | 49.1±1.4 | 33.5±1.4 |
| ECC (Simonovsky & Komodakis, 2017) | 76.1 | NA | NA | 76.8 | 75.0 | NA | NA |
| DGK (Yanardag & Vishwanathan, 2015) | 87.4±2.7 | 60.1±2.6 | 75.7±0.5 | 80.3±0.5 | 80.3±0.3 | 67.0±0.6 | 44.5±0.5 |
| DiffPool (Ying et al., 2018) | NA | NA | **78.1** | NA | NA | NA | NA |
| CCN (Kondor et al., 2018) | 91.6±7.2 | **70.6±7.0** | NA | 76.3±4.1 | 75.5±3.4 | NA | NA |
| IGN (Maron et al., 2018) | 83.9±13.0 | 58.5±6.9 | 76.6±5.5 | 74.3±2.7 | 72.8±1.5 | 72.0±5.5 | 48.7±3.4 |
| GIN (Xu et al., 2019) | 89.4±5.6 | 64.6±7.0 | 76.2±2.8 | 82.7±1.7 | NA | 75.1±5.1 | 52.3±2.8 |
| 1-2-3 GNN (Morris et al., 2019b) | 86.1 | 60.9 | 75.5 | 76.2 | NA | 74.2 | 49.5 |
| PPGN v1 (Maron et al., 2019) | 90.5±8.7 | 66.2±6.5 | 77.2±4.7 | 83.2±1.1 | 81.8±1.9 | 72.6±4.9 | 50±3.2 |
| PPGN v2 (Maron et al., 2019) | 88.9±7.4 | 64.7±7.5 | 76.4±5.0 | 81.2±2.1 | 81.8±1.3 | 72.2±4.3 | 44.7±7.9 |
| PPGN v3 (Maron et al., 2019) | 89.4±8.1 | 62.9±7.0 | 76.7±5.6 | 81.0±1.9 | 82.2±1.4 | 73±5.8 | 50.5±3.6 |
| LNGN (GCN) (de Haan et al., 2020) | 89.4±1.6 | 66.8±1.8 | 71.7±1.0 | 82.7±1.4 | 83.0±1.9 | 74.8±2.0 | 51.3±1.5 |
| GSN-e (Bouritsas et al., 2020) | 90.6±7.5 | 68.2±7.2 | 76.6±5.0 | 83.5±2.3 | NA | **77.8±3.3** | **54.3±3.3** |
| GSN-v (Bouritsas et al., 2020) | 92.2±7.5 | 67.4±5.7 | 74.6±5.0 | 83.5±2.0 | NA | 76.8±2.0 | 52.6±3.6 |
| SIN (Bodnar et al., 2021b) | NA | NA | 76.5±3.4 | 82.8±2.2 | NA | 75.6±3.2 | 52.5±3.0 |
| CIN (Bodnar et al., 2021a) | **92.7±6.1** | 68.2±5.6 | 77.0±4.3 | 83.6±1.4 | **84.0±1.6** | 75.6±3.7 | 52.7±3.1 |
| LPEGN | 89.5±6.1 | 70.0±11.3 | 74.5±2.3 | **83.7±1.5** | 83.2±0.8 | 74.3±3.7 | 47.9±3.0 |
| Best Rank | 6th | 2nd | 15th | 1st | 2nd | 7th | 10th |

## 6.2 SCALABILITY

We compare global permutation equivariant models with our local permutation equivariant model to assess the improvements in scalability offered by local permutation equivariance. Here we compare the GPU memory required by the model against the average size of graph in the dataset. It is expected that as the computational cost of global methods scales superlinearly with the size of the graph, due to the requirement to treat the entire graph as a single adjacency tensor, that local equivariance will have a lower computational cost as each update only requires local node neighbourhoods to be expressed as adjacency tensors, which are typically much smaller than the size of the graph. Therefore global methods scale with $\mathcal{O}(n^2)$, for graphs with $n$ nodes, while local methods scale with $\mathcal{O}(nm^2)$, where $m$ is the number of nodes in a node neighbourhood and typically $m \ll n$. Figure 4 shows how global and local permutation equivariant models scale with GPU memory usage as the average size of the graphs in the dataset increases. This will allow the LPEGN method to scale to graph datasets that was not possible with global equivariance.

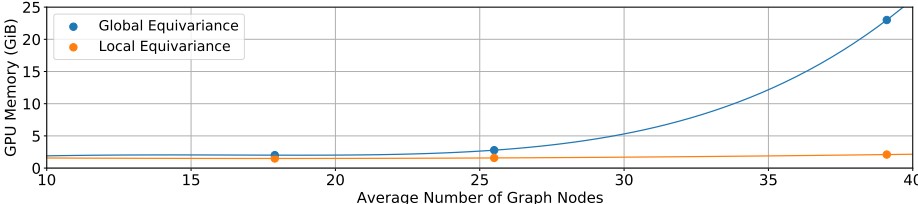

Figure 4: Computational cost of global and local permutation equivariant models with the same number of model parameters for datasets with varying average size graphs. For the local equivariance case local neighbourhoods were computed using 1-hop neighbourhoods.

## 7 FUTURE WORK

From Table 1 it is clear that IMDB-M is a dataset for which our method has weaker performance. As stated in Section A.3 between hidden local equivariant graph neural network layers for the experiments in this paper we only make use of order 1 and 2 representations. As it was shown by Maron et al. (2019) that increasing the order of the permutation representation increases the expressivity inline with the $k$-WL test, the expressivity of our method could be improved through the consideration of higher order permutation representations. Making use of higher order representations, we believe, would improve results on the IMBD-M dataset and therefore makes for an interesting future direction.

## 8 CONCLUSION

We present a graph neural network framework for building models comprising of local permutation equivariant update functions. The method presented is general in that it presents a framework for operating on sub-graphs with permutation equivariant convolutions, where a choice of representation space can be made depending on the expressivity required. This maintains expressivity in the update functions by utilising restricted representations, while improving scalability over global permutation equivariant methods by operating on smaller sub-graphs. We show that this method includes many previous published approaches as specific cases. Using a general approach as our framework does makes it easier to build provably expressive graph neural networks without the need to embed structural information about the task at hand, as is done in other methods. Further, we experimentally validate the method using $k = 1$ to create the sub-graphs and $\rho_1 \oplus \rho_2$ representations for the local update functions on a set of graph classification datasets. This model produces state-of-the-art results on one of the datasets, achieves second best results on two datasets, and is competitive on the remaining four. In addition, ranking the model against existing methods on each dataset shows that our method is one of the strongest performing methods. Furthermore, when compared to global permutation equivariant models our method offers a significant improvement in terms of the GPU memory usage, improving the scalability of the method.

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

## A APPENDIX

### A.1 ISOMORPHIC GRAPHS

An example of two isomporhic and two non-isomorphic graphs are shown in Figure 5. To a permutation invariant message passing update function utilising a max pooling aggregation function the isomorphic and non-isomorphic graphs are equivalent when updating the central node.

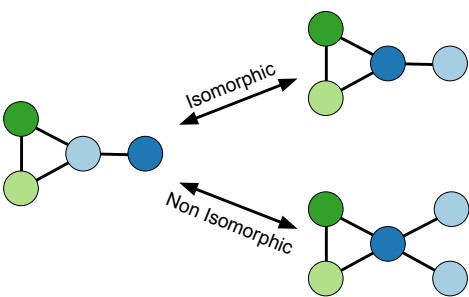

Figure 5: The initial graph on the left is isomorphic to the top graph on the right. On the other hand, the initial graph on the left is non-isomorphic to the bottom graph on the right. This is because there is no structure preserving mapping that has an inverse between the two graphs.

### A.2 MATHEMATICAL BACKGROUND

**Definition 5** *A group is a set $G$ with a binary operation $\circ$ satisfying the following laws:*
*(G0) (**Closure law**): For all $g, h \in G$, $g \circ h \in G$*
*(G1) (**Associative law**): $g \circ (h \circ k) = (g \circ h) \circ k$ for all $g, h, k \in G$*
*(G2) (**Identity law**): There exists $e \in G$ such that $g \circ e = e \circ g = g$ for all $g \in G$*
*(G3) (**Inverse law**): For all $g \in G$, there exists $h \in G$ with $g \circ h = h \circ g = e$*

**Definition 6** *A representation of a finite group on a finite-dimensional complex vector space $V$ is a homomorphism $\rho \to \mathrm{GL}(V)$ of the group to automorphisms of $V$ (Fulton & Harris, 2013). This allows group elements to be expressed as invertible matrices and the group operation to be matrix multiplication.*

### A.3 MODEL ARCHITECTURE

We consider the input graphs as an input feature space that is an order 2 representation. For each local permutation equivariant linear layer we use order 1 and 2 representations as the feature spaces. This allows for projection down from graph to node feature spaces through the basis for $\rho_2 \to \rho_1$, projection up from node to graph feature spaces through the basis for $\rho_1 \to \rho_2$, and mappings across the same order representations through $\rho_2 \to \rho_2$ and $\rho_1 \to \rho_1$. The final local permutation equivariant linear layer maps to order 0 representations through $\rho_2 \to \rho_0$ and $\rho_1 \to \rho_0$ for the task of graph level classification. In addition to the graph layers, we also add 3 MLP layers to the end of the model.

Despite these specific choices which were made to provide a baseline of our method for comparison to existing methods the framework we present is much more general and different representation spaces can be chosen. We present the general framework in Figure 6. This shows how different permutation representation spaces, $\rho_1 \oplus \rho2 \oplus \cdots \oplus \rho_i$, can be chosen for different layers in the model and how different $k$ values can be chosen when creating the sub-graphs in each layer.

Input Graph
$\rho_j$ - representations

$k$-hop Subgraph
$\rho_j$ - representations

Subgraph Convolution
$\rho_j$ -representations $\to \rho_1 \oplus \rho_2 \oplus \cdots \oplus \rho_i$ -representations

Reinsert Subgraphs
$\rho_1 \oplus \rho_2 \oplus \cdots \oplus \rho_i$ - representations

$k$-hop Subgraphs
$\rho_1 \oplus \rho_2 \oplus \cdots \oplus \rho_i$ - representations

Subgraph Convolution
$\rho_1 \oplus \rho_2 \oplus \cdots \oplus \rho_i$ - representations $\to \rho_1 \oplus \rho_2 \oplus \cdots \oplus \rho_i$ - representations

Reinsert Subgraphs
$\rho_1 \oplus \rho_2 \oplus \cdots \oplus \rho_i$ - representations

Stack Multiple Layers

$k$-hop Subgraphs
$\rho_1 \oplus \rho_2 \oplus \cdots \oplus \rho_i$ - representations

Subgraph Convolution
$\rho_1 \oplus \rho_2 \oplus \cdots \oplus \rho_i$ - representations $\to \rho_0$ - representations

Reinsert Subgraphs
$\rho_0$ - representations

Figure 6: The general framework for building models with permutation equivariance through representations of the permutation group is presented. In general for work on graphs the input representation $\rho_j = \rho_2$, but this could vary if the input was a set or other data structure. Then for each layer a choice of $k$ value can be made which is the number of hops to consider when extracting subgraphs. Also, for each layer the permutation representation type can be chosen where the choice of $\rho_1 \oplus \rho2 \oplus \cdots \oplus \rho_i$ is the order of representations required for that layer. Choosing larger order representations increases the expressivity of the model, although this increases the bases space increasing the computational cost.

## A.4 EXPRESSIVITY

Figure 7 shows the training accuracy achived by the LPEGN model across a range of datasets. (a, b, c, and d) show that the LPEGN model is able to achieve 100% training accuracy on PTC, Proteins, NCI1, and NCI109 datasets. This demonstrates that the model utilising only order 1- and 2-permutation representations is sufficiently expressive. In addition, (e) shows that the model achieves very close to 100% accuracy on the IMDBB dataset. On the other hand, (f) shows that the model training accuracy plateaus above 70% accuracy for the IMDBM dataset. This highlights the model is not sufficiently expressive to achieve 100% accuracy on this datset. As discussed in Section 7 we belive that utilising higher order permutation representations would make the model more expressive and as a result achieve a higher accuracy on IMDBM.

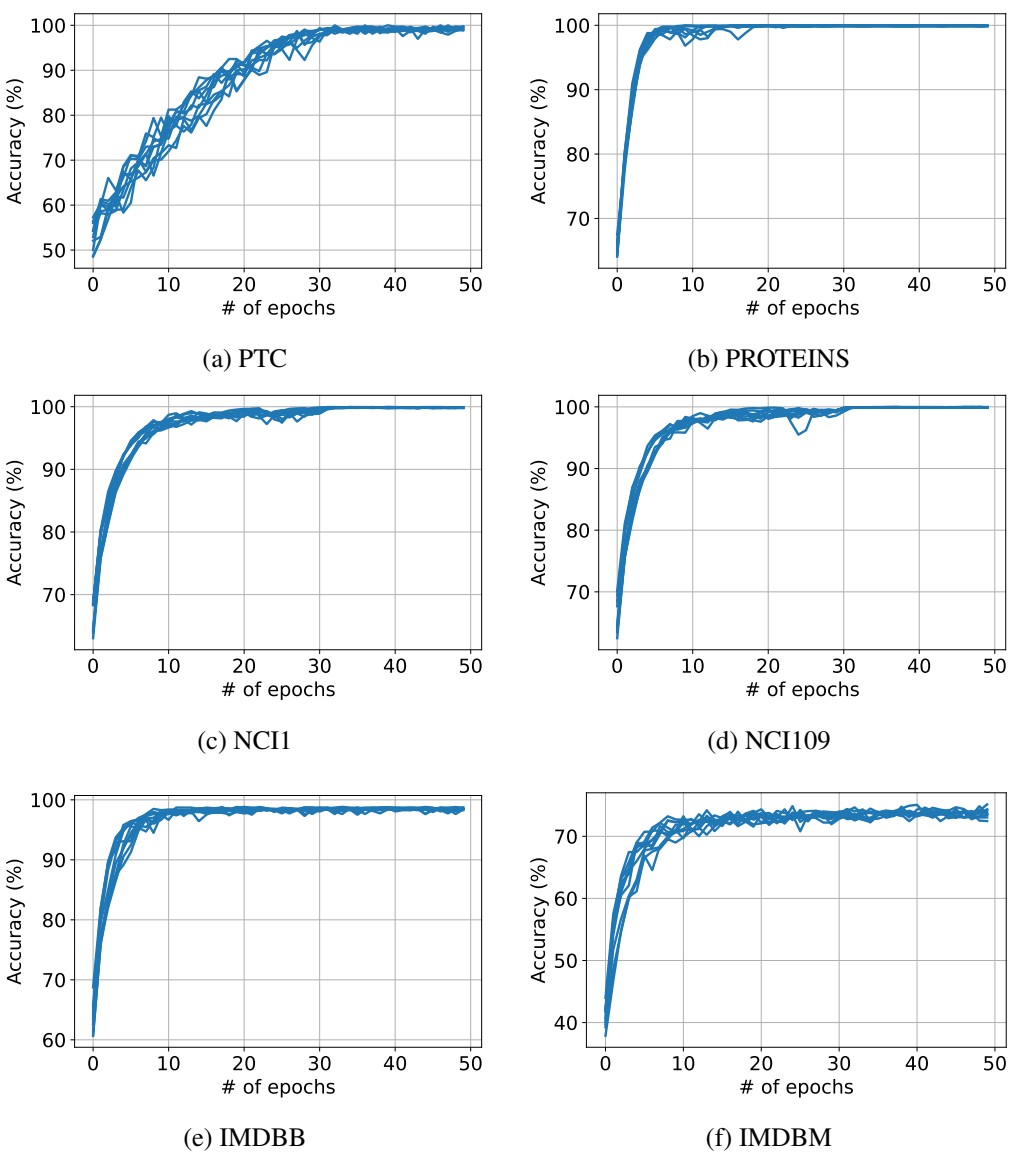

Figure 7: A visualisation of the training accuracy for benchmark graph classification datasets across 10-folds. Each plot shows the training accuracy as a percentage, where a higher percentage is better.

### A.5 COMPARISON OF RESULTS

In addition to the comparison across datasets in Table 1 Figure 8 shows the training accuracy distribution of the LPEGN method and compares to other methods from Table 1. The multimodal distribution of LPEGN for the PTC dataset highlights why it has a large standard deviation. This is likely a result of the fact that the PTC dataset is very small. Given the poor ranking of the LPEGN method in Table 1, comparing the results to other methods here highlights that the LPEGNN result is competitive. For the NCI1 and NCI109 datasets the distribution of results of our method highlight the strong performance of the method. For IMDBB and IMDBM the distribution of results for the LPEGN method also highlight that it is competative on these datasets.

Further, we propose an additional method of comparison, namely the counts of wins of our LPEGN method with other methods and the counts of significant wins. The result of comparing the counts of wins shown in Figure 9 highlights that our method is one of the strongest performing across the range of datasets. Where LPEGN under-performs against other methods this can largely be attributed to weaknesses on the PROTEINS and IMDBM datasets, which we suspect using higher order representations could improve.

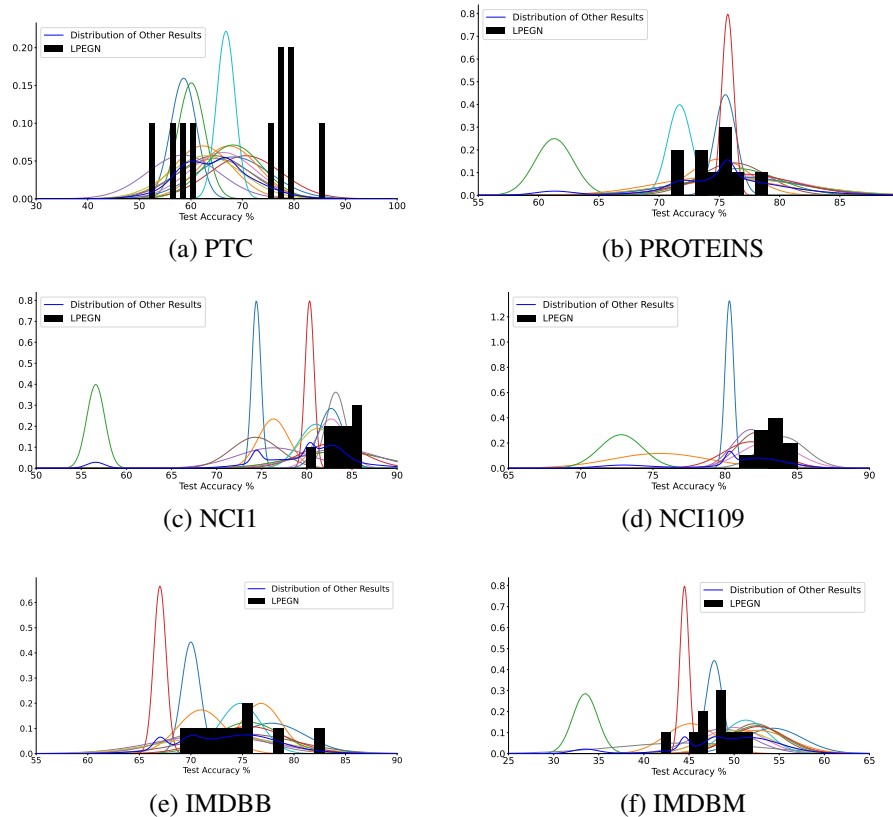

Figure 8: Comparison between out LPEGN method and other methods on benchmark graph classification datasets. Results for the LPEGNN method are presented as a histogram of the 10-fold runs. Each other method is given as a Gaussian distribution with mean and standard deviation as is presented in Table 1. In addition, we display the sum of the distributions for all other methods to show how our method compares.

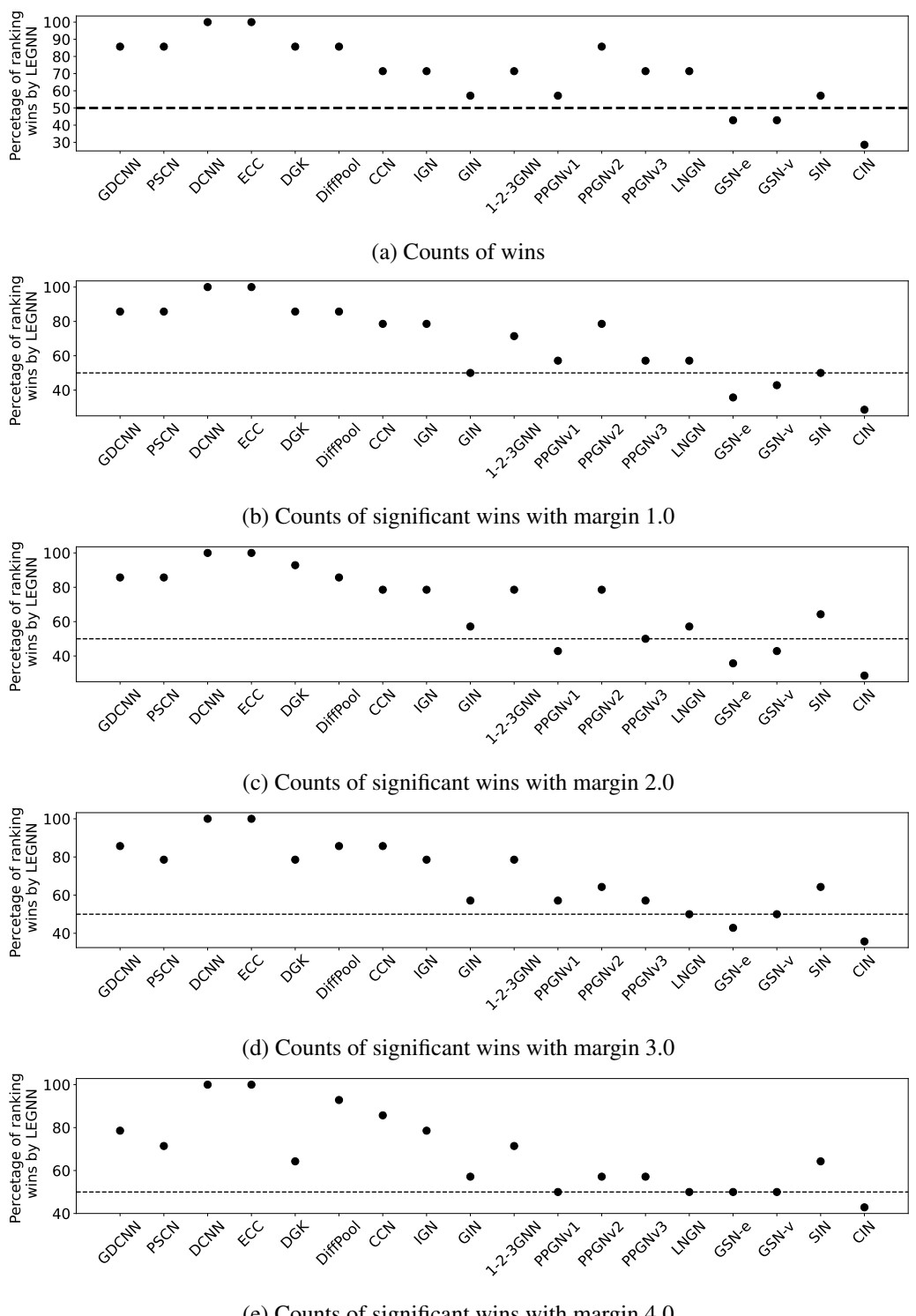

(a) Counts of wins

(b) Counts of significant wins with margin 1.0

(c) Counts of significant wins with margin 2.0

(d) Counts of significant wins with margin 3.0

(e) Counts of significant wins with margin 4.0

Figure 9: Presented is the percentage of ranking wins across the seven datasets for the LPEGN. A results above 50% means the LPEGN method beats the other method across the majority of datasets.

### A.6 CHOICE OF LOCAL NEIGHBOURHOOD

We show how this choice of $k$ value will impact the method through analysing the MUTAG dataset and comparing the size of sub-graphs found for different $k$ values, ranging from the most local, $k = 1$, up to equivalence of a global update, $k = 15$, shown in Figure 10.

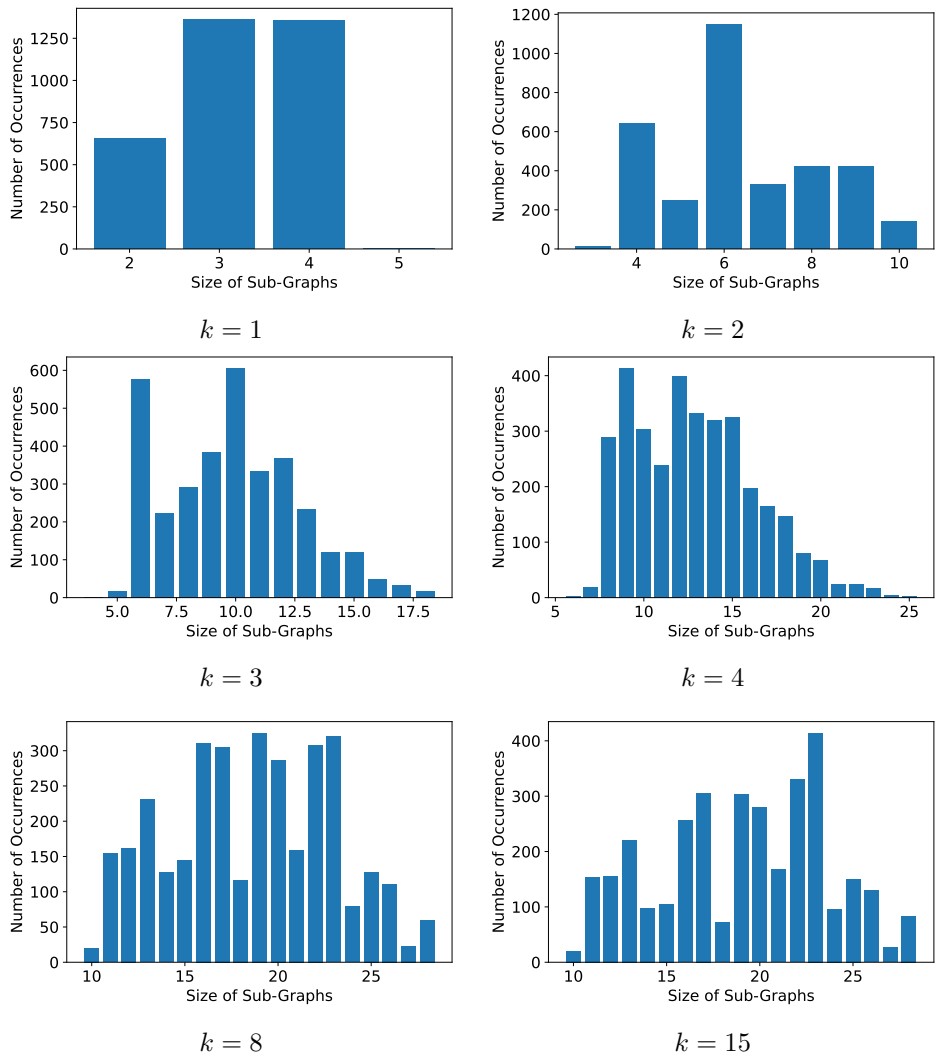

Figure 10: Histograms showing the number of occurrences of each size sub-graph in the MUTAG dataset for different choices of $k$ value, where $k$ is the number of hops to take away from the central node when selecting nodes to be included within a sub-graph.

## A.7 COMPARISON OF LOCAL AND GLOBAL FEATURE SPACES

We compare the case of global permutation equivariance to our local permutation equivariance, demonstrating how sub-graphs and the choice of representation is made in Figure 11.

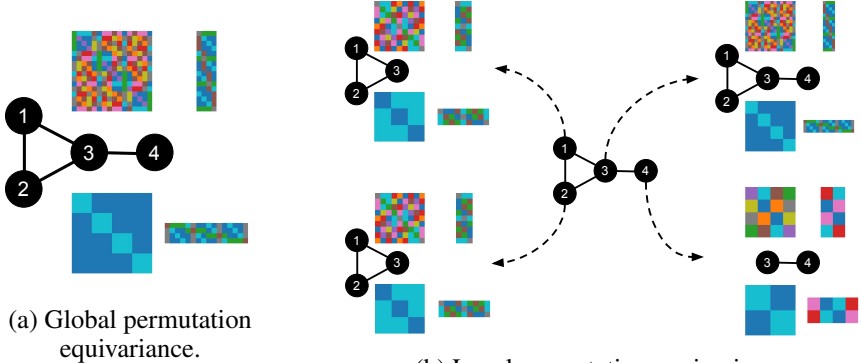

(a) Global permutation
equivariance.

(b) Local permutation equivariance.

Figure 11: (a) Global permutation equivariance and some of the possible representation choices. (b) Local permutation equivariance demonstrated through local neighbourhoods selected based on graph connectivity. Permutation equivariant representations are driven by the degree of the central node. Some of the possible representation choices are shown for each neighbourhood.

## A.8 PROOF OF NO LOSS OF EXPRESSIVITY WHEN USING RESTRICTED REPRESENTATIONS

Restricting the permutation representation, $\rho_n$, from a group $G$ with $n$ nodes to a subgroup $H$ with $m$ nodes yields the restricted representation $\tilde{\rho}_m := \text{Res}_H^G(\rho_n)$. The bases for the permutation representation $\rho_n$ from a set of nodes to a set of nodes is given in Figure 3 and has 2 basis elements. We show that the restricted representation adds 3 more basis elements in Figure 3. From definition 2 there are no edge features associated between the nodes in the sub-graph and the nodes outside of the sub-graph. Therefore 2 of the basis elements introduced in the restricted representations are always multiplied by zeros and not required. Further, the extra $3^{\text{rd}}$ basis element introduced is simply weighting the node features not part of the sub-graph by themselves and as our method extracts a sub-graph for each node in the graph this update is subsumed by the sub-graph update of that node. Therefore the restricted representation for our framework is equal to the permutation representation of a lower dimensional space and $\tilde{\rho}_m = \rho_m$. Therefore, the proof of no loss of expressivity when using restricted representations follows from the proof that $k$-order graph networks are as powerful as $k$-WL (Maron et al., 2019).

## A.9 IMPLEMENTING OTHER MODELS WITHIN OUR FRAMEWORK

We have re-drawn our model in a step-by-step format to try and highlight the difference to other models and make clear that this is a more general framework for learning permutation equivariant models in Figure 6. In the datasets used, for graph classification benchmark tasks, the input to the model is a graph with node and edge features, this can be represented as $2^{\text{nd}}$ order permutation representation, so the input representation would be $j = 2$. The convolution can then map from this representation, $\rho_j$, to multiple different representation spaces, $\rho_0 \oplus \rho_1 \oplus \cdots \oplus \rho_i$. Subsequent convolutions can then map from these multiple permutation representations, $\rho_0 \oplus \rho_1 \oplus \cdots \oplus \rho_i$, to multiple different permutation representations, $\rho_0 \oplus \rho_1 \oplus \cdots \oplus \rho_i$. The choice of representations used can be made depending on a trade off between expressivity and computational cost, as lower order representation spaces have less expressivity, but also lower computational cost.

**Local Natural Graph Networks** (LNGNs) (de Haan et al., 2020) take the input feature space and embed this into an invariant scalar feature of the edge neighbourhood graph. This is the same as using specific choice $k$-hop sub-graph creation and permutation representation space for the sub-graph convolution. In the case of LNGNs the choice would be $k = 1$ and mapping the input feature space to representation $\rho_0$ creating a permutation invariant feature space. Then any graph neural

network with invariant features can be used, in the paper the choice made is to use a GCN (Kipf & Welling, 2016), which can also be covered by our framework. Here the choice would again be to use $k = 1$ when creating the subgroups and using a subgraph convolution with representation spaces $\rho_0 \to \rho_0$.

**Global Equivariant Graph Networks** (EGNs) (Maron et al., 2018) use a choice of $k = n$, for $n$-node graphs when creating the sub graphs, which corresponds to not selecting a sub graph and instead operating over the entire graph. They then use the representation space $\rho_2 \to \rho_2$ mapping from a graph feature space to a graph feature space.

**Local Permutation Equivariant Graph Networks** (LPEGN) (Ours) In our paper we choose to use $k = 1$ throughout to keep inline with the vast majority of previous work on graph neural networks, but we use a representation space of $\rho_1 \oplus \rho_2 \to \rho_1 \oplus \rho_2$ in the hidden layers of the model and we note that this was simply a choice that seemed a simple case to present as a comparison with previous work in the benchmark classification task.

