# OpenReview forum: "Local Permutation Equivariance For Graph Neural Networks"
_ICLR.cc/2022/Conference — ICLR 2022 Submitted_

### Official Review · Reviewer_zCkV · 2021-11-02

**Correctness:** 2
**Technical Novelty And Significance:** 2
**Empirical Novelty And Significance:** Not applicable
**Recommendation:** 3
**Confidence:** 4

**Details Of Ethics Concerns:**

-

**Main Review:**

**Strengths**

1. **Scalability** - the proposed framework introduces a scalable version of global equivariant graph networks [Maron et.al. 2018].
2. **Performance** - on the selected datasets the model has been evaluated on, the proposed model performs relatively well across all datasets.



**Weaknesses**

1. **Motivation** - the motivation for the proposed framework is a bit unclear to me. It seems like a less restricted instantiation of local natural graph networks which choses to work with equivariant layers instead of message passing layers in the local updates, but the claims and justifications feel rather weak and unsupported.
2. **Theory** - the paper claims that the framework maintains expressivity or even can achieve improved expressivity (Section 5), but it is not clear over what other models? And there are no theoretical results which show that, other then stating it several times in the paper.



**Clarifications**

1. **Applicability to new graphs** - how does the network handle subgraphs of unseen sizes during training?
2. **Framework description** - I find it confusing that the framework is only illustrated in a figure which is not close to the text describing it.



**Summary Of The Paper:**

The paper introduces the framework of locally permutation equivariant graph neural networks.
This framework applies permutation equivariant layers [Maron et. al. 2018] to local node neighborhoods by treating them as separate subgraphs and using a weight sharing scheme for subgraphs of the same size.
The authors build their framework by discussing the different choices made -- local neighborhoods, weight sharing, and representation space.
The authors also provide a category theory point of view of their framework.

**Summary Of The Review:**

I think this paper introduces an interesting framework but fails to justify and motivate its constructions. I further feel that the theoretical analysis of the framework is lacking.

---

> ### Author Response · Authors · 2021-11-12
> **Response to reviewer concerns**
>
> We have added in more details showing how our method provides a general framework for building permutation equivariant models and how other methods can be expressed as specific cases of our architecture in section A.9. We have also added a proof that there is no loss of expressivity using the restricted representations which our model uses in section A.8. We have also added more information on the framework we present and how models can be constructed from this including the values we choose for the benchmark we present in the paper in section A3. We hope these make the contribution of this paper clear.
>
> We have re-drawn our model in a step-by-step format to try and highlight the difference from other models and make clear that this is a more general framework for learning permutation equivariant models, see section A3 in the updated paper. Generally speaking the convolution in our framwork can then map from multiple permutation representations $\rho_{1} \oplus \rho{2} \oplus \dots \oplus \rho_{i}$ to multiple different permutation representations $\rho_{1} \oplus \rho{2} \oplus \dots \oplus \rho_{i}$. The choice of representations used can be made depending on a trade off between expressivity and computational cost, as lower order representation spaces have less expressivity, but also lower computational cost.
>
> Local Natural Graph Networks (LNGNs) [1] take the input feature space and embed this into an invariant scalar feature of the edge neighbourhood graph. This is the same as using specific choice $k$-hop subgraph creation and permutation representation space for the subgraph convolution. In the case of LNGNs the choice would be $k=1$ and mapping the input feature space to representation $\rho_{0}$ creating a permutation invariant feature space. Then they use any graph neural network with invariant features, in the paper the choice made is to use a GCN, which can also be covered by our framework. Here the choice would again be to use $k=1$ when creating the subgroups and using a subgraph convolution with representation spaces $\rho_{0} \rightarrow \rho_{0}$. In our paper we choose to use $k=1$ throughout to keep inline with the vast majority of previous work on graph neural networks, but we use a representation space of $\rho_{1} \oplus \rho_{2}$ in the hidden layers of the model and we note that this was simply a choice that seemed a simple case to present as a comparison with previous work in the benchmark classification task.
>
> Proof of no loss of expressivity when using restricted representations. Restricting the permutation representation, from a group G with n nodes to a subgroup H with m nodes yields the restricted representation. The bases for the permutation representation from a set of nodes to a set of nodes is given in Figure 3 Section 5.1 and has 2 basis elements. We show in Section 5.1 that the restricted representation adds 3 more basis elements and we show the bases in Figure 3. From the definition of selecting a subgroup, definition 2 in the paper, there are no edge features associated between the nodes in the subgraph and the nodes outside of the subgraph. Therefore 2 of the basis elements introduced in the restricted representations are always multiplied by zeros. Further, the extra 3rd basis element introduced is simply weighting the node features not part of the subgraph by themselves and as our method extracts a subgraph for each node in the graph this update is subsumed by the subgraph update of that node. Therefore the restricted representation for our framework is equal to the permutation representation of a lower dimensional space and. Therefore, the proof of no loss of expressivity when using restricted representations follows from the proof that k-order graph networks are as powerful as k-WL [5]. We have added this into the appendix of the paper.
>
> For the question on handling subgraphs of unseen size during training. The weight sharing is a choice in the framework and it is not required that each subgroup of differing degree is updated with separate weights. One option is to choose that each subgroup of differing degree is updated with separate weights. On the other hand, if some subgroup degrees have very low occurrence or the method is required to generalise to subgraph sizes out-side of the training set then the weight sharing could be chosen to reflect that, for example the weight sharing could be chosen to be 5, 6-10, 11-15, and 16, which would ensure every subgraph size is captured.
>
> We have added an updated version of the paper to address the comments by reviewers so please see that for further details. Specifically see an updated abstract, end of introduction, and conclusion to clarify the contributions. Further, section 2 and 4 have been updated to include comparisons to more methods. Finally, Appendix sections A3,5,8,9 have been added or updated to further make clear our contributions.
>
> [1] Natural graph networks.
>
> [5] Provably powerful graph networks.

---

> > ### Comment · Reviewer_zCkV · 2021-11-29
> > **Thank you for your response**
> >
> > I thank the authors for their response.
> >
> > After reading your answers, the motivation for the paper is still unclear to me, I think there is more work to do in the presentation of the paper.
> >
> > Further, regarding the "no loss of expressive power" claim, I am not sure I understand your proof. I am not sure how the fact that the restricted representation is equivalent to the permutation representation on a subgroup implies conservation of expressive power on the entire graph.
> >
> > I appreciate your work and effort, but I feel that this paper is not ready for publication and I therefore keep my score.

---

### Official Review · Reviewer_KECi · 2021-11-02

**Correctness:** 3
**Technical Novelty And Significance:** 3
**Empirical Novelty And Significance:** 3
**Recommendation:** 5
**Confidence:** 2

**Main Review:**

The proposed method is impressively practical in terms of performance and efficiency.  The proposed local permutation equivariant local update function is more GPU memory efficient than the global counterpart. Moreover, this work achieves impressive performance in 7 graph classification networks.

However, I have several concerns about the paper which I will detail below:

- While I appreciate the paper’s providing sufficient background about graph neural networks to improve readiness,  I am not sure I clearly understand the relation of the provided background with the proposed method. For example, in section 3, I can understand the mechanism of global equivariance. But  I am not sure how section 3 can help elaborate the method. I probably would recommend moving part of those stuff into supplementary.

- I wonder about the novelty compared with other works in Graph convolution. From my viewpoint, the Graph Conv/ Messaging pass network is also locally permutation-equivariant. To be more specific, LNGN is a method that shares the similar idea of using locally permutation-equivariant update functions. It also shows competing performance in table 1 of the paper. Is this work essentially the extension of LNGN?

- Missing baselines GPU memory efficiency comparison. I believe other graph neural networks that employ local update functions also benefit from GPU memory efficiency. I thus recommend the paper also provide a comparison with those methods.

- Experiments for the dataset that has the well-defined local symmetry -- say mesh data. I understand that this work aims to handle natural graph data. But it would be also very interesting to check out the performance of the proposed local permutation equivariant update function in mesh data. Because I think it might be helpful to validate the expressiveness of proposed functions.


**Summary Of The Paper:**

This paper introduces the local permutation equivariant network (LPEGN). Specifically, this work proposes to apply permutation equivariant update functions locally --- i.e., operating in a local neighborhood. The benefit of doing so is that the large graph is handled through sub-graph, which saves a lot of GPU memory. Moreover, this work handled different sizes of neighborhoods by proposing a heterogeneous weight-sharing mechanism. Weight is shared w.r.t neighborhood size -- local neighborhoods with the same size share weights. While being flexible to input sizes, the local update function is also expressive. The paper demonstrates its superiority in several graph classification benchmarks.

**Summary Of The Review:**

While the proposed method is practical, the novelty of methods is limited when compared with existing works that also employ locally permutation equivariant update functions. More importantly, the paper didn't provide a clear comparison to make the proposed method stand out from the previous related works. Thus, I currently vote for weakly reject. However, I still would like to hear more from the authors if I have any misunderstandings.

---

> ### Author Response · Authors · 2021-11-12
> **Response to reviewer concerns**
>
> With regards to a lack of clarity around the background section we have made updates to this alongside many other sections of the paper to improve this.
>
> It is not true that Graph/Conv/ Message passing is locally permutation-equivariant. Message passing is permutation equivariant to global permutations of the graph, but each local update on sub-graphs is permutation invariant. This has previously been proved to limit the expressivity of message passing models with a permutation invariant update function.
>
> We have re-drawn our model in a step-by-step format to try and highlight the difference from other models and make clear that this is a more general framework for learning permutation equivariant models, see section A3 in the updated paper. Generally speaking the convolution in our framwork can then map from multiple permutation representations $\rho_{1} \oplus \rho{2} \oplus \dots \oplus \rho_{i}$ to multiple different permutation representations $\rho_{1} \oplus \rho{2} \oplus \dots \oplus \rho_{i}$. The choice of representations used can be made depending on a trade off between expressivity and computational cost, as lower order representation spaces have less expressivity, but also lower computational cost.
>
> Local Natural Graph Networks (LNGNs) [1] take the input feature space and embed this into an invariant scalar feature of the edge neighbourhood graph. This is the same as using specific choice $k$-hop subgraph creation and permutation representation space for the subgraph convolution. In the case of LNGNs the choice would be $k=1$ and mapping the input feature space to representation $\rho_{0}$ creating a permutation invariant feature space. Then they use any graph neural network with invariant features, in the paper the choice made is to use a GCN, which can also be covered by our framework. Here the choice would again be to use $k=1$ when creating the subgroups and using a subgraph convolution with representation spaces $\rho_{0} \rightarrow \rho_{0}$. In our paper we choose to use $k=1$ throughout to keep inline with the vast majority of previous work on graph neural networks, but we use a representation space of $\rho_{1} \oplus \rho_{2}$ in the hidden layers of the model and we note that this was simply a choice that seemed a simple case to present as a comparison with previous work in the benchmark classification task.
>
> The section on scalability shows how our local permutation equivariant method compares to global permutation equivariance [2]. This is due to our method maintaining the expressivity of global permutation equivariance as both methods are as powerful as the k-WL test when using k-order representations. It has been acknowledged by many works that global permutation equivariant models scale badly and therefore we aimed to maintain the expressivity while improving the scalability. We show how both global and local permutation methods scale in terms of GPU memory usage for varying graph size. We limit the experiment to graphs of size 40 as the global method runs out of memory on a single Titan RTX GPU with 24GB of memory if the graphs get any larger, while our method scales better with graph size.  The datasets used for the scalability analysis were MUTAG, PTC, and PROTEINS from the benchmark classification tasks as these have an increasing average sized graph in the dataset. We state in section 6.2 on scalability that global permutation models scale with $\mathcal{O}(n^{2})$ for $n$ nodes, while our method scales with $\mathcal{O}(nm^{2})$ where $m$ is the size of the subgraph and in general $m \ll n$. We did not include a comparison in the plot to a local update function such as a message passing neural network with permutation invariant update function, as these do not have the same expressivity as the other two models, although it is known they have improved scalability as they scale linearly i.e. with $\mathcal{O}(n)$. Although the improved scalability is not helpful if the method has less expressivity.
>
> We note that some other papers do add in simple toy examples to showcase the benefit of the approach, but we feel that for permutation equivariant representations this is already well covered in [2]. Further, making a specific toy example where we know our method will outperform the method we compare to adds little value and instead we compare to a range of leading methods on a widely used set of graph benchmark tasks.
>
> We have added an updated version of the paper to address the comments by reviewers so please see that for further details. Specifically see an updated abstract, end of introduction, and conclusion to clarify the contributions. Further, section 2 and 4 have been updated to include comparisons to more methods. Finally, Appendix sections A3,5,8,9 have been added or updated to further make clear our contributions.
>
> [1] Natural graph networks.
>
> [2] Invariant and equivariant graph networks.

---

### Official Review · Reviewer_Y3RZ · 2021-11-02

**Correctness:** 2
**Technical Novelty And Significance:** 2
**Empirical Novelty And Significance:** 2
**Recommendation:** 3
**Confidence:** 4

**Main Review:**

Initial Recommendation: Rejection

Reason: In my view, in current format, the weaknesses outweigh the strengths of the paper. Please see details below.

Strengths:

1. Idea - Use of restricted representation of symmetric groups to reduce dimension of vector space - associated with group representation for linear permutation equivariant layers.
2. Practicality - Employ a weight sharing scheme when the size of the node neighborhood is the same
3. Demonstrate scalability in terms of GPU memory usage

Weakness:

For me, the main weakness of this paper is in its exposition and clarity that raised several questions. I have listed my main concerns below.
1.  Lack of precision and clarity in the paper, for e.g. (i) definition 2 is never again used in the paper (ii) section 3.1 The space of linear equivariant layers, given by bell number (4)) is 15 - what do you mean reducing linear layer to just 15 parameters?
2. The authors claim that there is no loss of expressivity while using restricted representations of finite symmetric groups but there is no formal proof for the same? Does this always hold? Do you need them to be normal subgroups? Is there any specific requirement for the representations (what happens when irreducible reps are used, etc) used?
3. While considering node neighborhood and edge neighborhood morphism misses out comparisons with relevant works which use subgraph counting isomorphism/ automorphism over local neighborhood [1][2][3][4] to obtain provably expressive representations The mentioned works also almost always perform better than the proposed work in the datasets.
4. The scalability studies are incomplete - e.g. do not describe what the datasets used are? how many graphs are there in the dataset? etc?
5. No study on effect on performance when k>1 (hops) is used (the plots in the appendix only show the size)

References:

1. Bouritsas, Giorgos, et al. "Improving graph neural network expressivity via subgraph isomorphism counting." arXiv preprint arXiv:2006.09252 (2020).
2. Morris, Christopher, Gaurav Rattan, and Petra Mutzel. "Weisfeiler and Leman go sparse: Towards scalable higher-order graph embeddings." arXiv preprint arXiv:1904.01543 (2019)
3. Bodnar, Cristian, et al. "Weisfeiler and lehman go cellular: Cw networks." arXiv preprint arXiv:2106.12575 (2021).
4. Bodnar, Cristian, et al. "Weisfeiler and lehman go topological: Message passing simplicial networks." arXiv preprint arXiv:2103.03212 (2021).

**Summary Of The Paper:**

Graph Neural Networks have recently become the state of the art for tasks on graphs due to their flexibility, scalability. In this work, the authors propose a framework to build GNN's that operate on local node neighborhoods in a permutation equivariant way - and argue that since LPEGN operates on lower dimensional spaces in comparison to regular GNN's the proposed technique offers significant improvements  in terms of GPU memory usage. The authors make use of category theory basics - and employ restricted representations of finite symmetric groups (i.e. fix some nodes while permuting other elements) based on the number of nodes in the neighborhood of the node - and ensure there is weight sharing between nodes with the same degree to achieve their objective.

**Summary Of The Review:**

The authors propose a practical framework mainly aimed to reduce GPU memory usage via the use of representation theory. However, for me, the main weakness of this paper is in its exposition and clarity - including lack of proofs for statements made, lack of comparison with appropriate baselines, incomplete analysis, etc,

---

> ### Author Response · Authors · 2021-11-12
> **Response to reviewer concerns**
>
> We would like to thank the reviewer for pointing out additional work on this area and it is an oversight on our behalf to miss these out of the comparison. We have now added these into the discussion in the paper.
>
> As we are considering graphs as concrete graphs, definition 2 provides a definition of a sub-concrete graph. We have added “For brevity we refer to sub-concrete graphs as subgraphs throughout the paper.” to make this clear.
>
> Maron et al [2] use a combinatorial approach to show that the space of permutation equivariant linear layers for graph structured data, in our terminology the representation space $\rho_{2}$, is of dimension equal to $2k^{\mathrm{th}}$ Bell number [4]. Therefore for $\rho_{2}$ this is equal to the 4th Bell number, which is 15. If no permutation equivariance was required we could simply use an MLP and flatten the graph, then the number of trainable parameters would be $n^{2} \times n^{2} = n^{4}$. This means that requiring permutation equivariance significantly reduces the number of trainable parameters  $n^{4} \ll 15$.
>
> For the proof of no loss of expressivity: Restricting the permutation representation, from a group G with n nodes to a subgroup H with m nodes yields the restricted representation. The bases for the permutation representation from a set of nodes to a set of nodes is given in Figure 3 Section 5.1 and has 2 basis elements. We show in Section 5.1 that the restricted representation adds 3 more basis elements and we show the bases in Figure 3. From the definition of selecting a subgroup, definition 2 in the paper, there are no edge features associated between the nodes in the subgraph and the nodes outside of the subgraph. Therefore 2 of the basis elements introduced in the restricted representations are always multiplied by zeros. Further, the extra 3rd basis element introduced is simply weighting the node features not part of the subgraph by themselves and as our method extracts a subgraph for each node in the graph this update is subsumed by the subgraph update of that node. Therefore the restricted representation for our framework is equal to the permutation representation of a lower dimensional space and. Therefore, the proof of no loss of expressivity when using restricted representations follows from the proof that k-order graph networks are as powerful as k-WL [5]. We have added this into the appendix of the paper.
>
> Need to be normal subgroups? No, the choice of subgraphs is given in definition 2.
> Requirements for representations? Irreps? We are considering the natural permutation representation of the group, the n-dimensional representation. Considering irreducible representations could be an interesting future direction.
> Didn’t consider k>1. To keep the computational cost of the experimental section to a minimum we only looked at k=1. It appears to perform well in the experiments and we didn’t have any specific intuition in the datasets to make a different choice. Clearly experimenting over a range of k values could be done in the future with adequate resources.
>
> The section on scalability shows how our local permutation equivariant method compares to global permutation equivariance [2]. This is due to our method maintaining the expressivity of global permutation equivariance as both methods are as powerful as the k-WL test when using k-order representations. It has been acknowledged by many works that global permutation equivariant models scale badly and therefore we aimed to maintain the expressivity while improving the scalability. We show how both global and local permutation methods scale in terms of GPU memory usage for varying graph size. We limit the experiment to graphs of size 40 as the global method runs out of memory on a single Titan RTX GPU with 24GB of memory if the graphs get any larger, while our method scales better with graph size.  The datasets used for the scalability analysis were MUTAG, PTC, and PROTEINS from the benchmark classification tasks as these have an increasing average sized graph in the dataset. We state in section 6.2 on scalability that global permutation models scale with $\mathcal{O}(n^{2})$ for $n$ nodes, while our method scales with $\mathcal{O}(nm^{2})$ where $m$ is the size of the subgraph and in general $m \ll n$.
>
> We have added an updated version of the paper to address the comments by reviewers so please see that for further details. Specifically see an updated abstract, end of introduction, and conclusion to clarify the contributions. Further, section 2 and 4 have been updated to include comparisons to more methods. Finally, Appendix sections A3,5,8,9 have been added or updated to further make clear our contributions.
>
> [1] Natural graph networks.
>
> [2] Invariant and equivariant graph networks.
>
> [4] https://en.wikipedia.org/wiki/Bell_number
>
> [5] Provably powerful graph networks.

---

### Official Review · Reviewer_auCg · 2021-11-04

**Correctness:** 2
**Technical Novelty And Significance:** 1
**Empirical Novelty And Significance:** 1
**Recommendation:** 3
**Confidence:** 3

**Main Review:**

The contribution of this paper is unclear to me. The approach is very similar with 'Natural graph networks' by Pim de Haan, Taco Cohen, Max Welling (NEURIPS 2020) and I would have like to see a clear comparison of the approach and the results.
The paper is very hard to follow. I agree that category theory is a very abstract field and an introduction is beyond the scope of the paper but it would have been nice to give some simple examples on toy examples highlighting the benefits of the approach proposed here.
It is impossible to understand the architecture used here as the description in Section 5 remains at a very high level. I could not find the code associated with this paper.
The experiment about scalability in section 6.2 is not convincing at all as it deals with graphs with a maximum average number of nodes of 40.

**Summary Of The Paper:**

This paper introduces local permutation equivariant graph network. The main motivation for introducing these graph neural networks is to improve in term of scalability with respect to global permutation equivariant models.
This paper uses the very abstract language of category theory. In Section 6, the authors provide some experimental results on some real-world graph classification problems.

**Summary Of The Review:**

The contribution of the paper is unclear. Provide more examples and convincing experiments about scalability which is one of your main motivation.

---

> ### Author Response · Authors · 2021-11-12
> **Response to reviewer concerns**
>
> We have re-drawn our model in a step-by-step format to try and highlight the difference from other models and make clear that this is a more general framework for learning permutation equivariant models, see section A3 in the updated paper. Generally speaking the convolution in our framwork can then map from multiple permutation representations $\rho_{1} \oplus \rho{2} \oplus \dots \oplus \rho_{i}$ to multiple different permutation representations $\rho_{1} \oplus \rho{2} \oplus \dots \oplus \rho_{i}$. The choice of representations used can be made depending on a trade off between expressivity and computational cost, as lower order representation spaces have less expressivity, but also lower computational cost.
>
> Local Natural Graph Networks (LNGNs) [1] take the input feature space and embed this into an invariant scalar feature of the edge neighbourhood graph. This is the same as using specific choice $k$-hop subgraph creation and permutation representation space for the subgraph convolution. In the case of LNGNs the choice would be $k=1$ and mapping the input feature space to representation $\rho_{0}$ creating a permutation invariant feature space. Then they use any graph neural network with invariant features, in the paper the choice made is to use a GCN, which can also be covered by our framework. Here the choice would again be to use $k=1$ when creating the subgroups and using a subgraph convolution with representation spaces $\rho_{0} \rightarrow \rho_{0}$. In our paper we choose to use a representation space of $\rho_{1} \oplus \rho_{2}$ in the hidden layers of the model and we note that this was simply a choice to compare with previous work.
>
> We have added in more details showing how our method provides a general framework for building permutation equivariant models and how other methods can be expressed as specific cases of our architecture in section A.9. We have also added a proof that there is no loss of expressivity using the restricted representations which our model uses in section A.8. We have also added more information on the framework we present and how models can be constructed from this including the values we choose for the benchmark we present in the paper in section A3. We hope these make the contribution of this paper clear.
>
> We have extended the explanation of the model architecture in Appendix A.3 and introduced a new figure to explain the architecture. We hope the addition of this helps make clear how the architecture works; what the main building blocks of the model are; and what choices can be made in terms of the subgraph creation and choice of permutation representation spaces.
>
> We note that some other papers do add in simple toy examples to showcase the benefit of the approach, but we feel that for permutation equivariant representations this is already well covered in [2].
>
> The section on scalability shows how our local permutation equivariant method compares to global permutation equivariance [2]. This is due to our method maintaining the expressivity of global permutation equivariance as both methods are as powerful as the k-WL test when using k-order representations. It has been acknowledged by many works that global permutation equivariant models scale badly and therefore we aimed to maintain the expressivity while improving the scalability. We show how both global and local permutation methods scale in terms of GPU memory usage for varying graph size. We limit the experiment to graphs of size 40 as the global method runs out of memory on a single Titan RTX GPU with 24GB of memory if the graphs get any larger, while our method scales better with graph size.  The datasets used for the scalability analysis were MUTAG, PTC, and PROTEINS from the benchmark classification tasks as these have an increasing average sized graph in the dataset. We state in section 6.2 on scalability that global permutation models scale with $\mathcal{O}(n^{2})$ for $n$ nodes, while our method scales with $\mathcal{O}(nm^{2})$ where $m$ is the size of the subgraph and in general $m \ll n$. We did not include a comparison in the plot to a local update function such as a message passing neural network with permutation invariant update function, as these do not have the same expressivity as the other two models, although it is known they have improved scalability as they scale linearly i.e. with $\mathcal{O}(n)$. Although the improved scalability is not helpful if the method has less expressivity.
>
> We have added an updated version of the paper to address the comments by reviewers so please see that for further details. Specifically see an updated abstract, end of introduction, and conclusion to clarify the contributions. Further, section 2 and 4 have been updated to include comparisons to more methods. Finally, Appendix sections A3,5,8,9 have been added or updated to further make clear our contributions.
>
> [1] Natural graph networks
>
> [2] Invariant and equivariant graph networks.

---

> > ### Comment · Reviewer_auCg · 2021-11-30
> > **Thank you for your answer**
> >
> > A minor comment, in [2] dealing with global permutation, experiments are done with graphs of size larger than 40.

---

> > > ### Author Response · Authors · 2021-12-07
> > > **Response to reviewer**
> > >
> > > This is true, [2] does test on the Collab dataset. Here we didn't test can the model run on a single graph of specific size, but could a reasonable training loop be conducted on the dataset. For the scalability experiment we tested on a single GPU and kept batch size, feature space size, etc. the same between both models so as to provide a fair comparison. Therefore, we are not trying to argue that it is not possible to run [2] on larger datasets, where it would be if more gpus are used or a lower batch size is used, but trying to argue that our method scales in terms of gpu memory usage to larger graphs better.
> > >
> > > I hope this helps clarify the choices made in the scalability experiments and I would be happy to add further details into the appendix to make this clear.

---

### Decision · Program_Chairs · 2022-01-20

**Decision:**

Reject

**Comment:**

This paper presents a graph neural network (GNN) architecture that adopts locally permutation-equivariant constructs, which has better scalability compared to globally permutation-equivariant GNNs, and the paper claims this change also does not lose expressivity of the network.  All reviewers unanimously recommended rejection, and the main issues are the clarity and writing, to the point where it becomes hard for a reader to follow the precise implementation of the proposed approach and how that compares to prior work.  Therefore in its current form this paper is not yet ready for publication at ICLR.  When the authors work toward the next revision I’d suggest clarifying a little more about the precise algorithmic implementation of the proposed ideas, with a bit of additional intuition from a higher-level, rather than staying at the current level of technicality.